# Integration of molecular networking and fingerprint analysis for studying constituents in Microctis Folium

Yang Bai[1,2]☯, Qiang Jia[3]☯, Weiwei Su[1,2], Zenghao Yan[1,2], Wenhui Situ[1,2], Xiang He[1,2], Wei Peng[1,2], Hongliang Yao[1,2]*

**1** Guangdong Engineering & Technology Research Center for Quality and Efficacy Reevaluation of Post-Market Traditional Chinese Medicine, School of Life Sciences, Sun Yat-sen University, Guangzhou, China, **2** Guangdong Key Laboratory of Plant Resources, School of Life Sciences, Sun Yat-sen University, Guangzhou, China, **3** Food Department, Guangzhou City Polytechnic, Guangzhou, China

☯ These authors contributed equally to this work.
* yaohliang@mail.sysu.edu.cn

**Data Availability Statement:** All relevant data are within the manuscript and its Supporting Information files.

**Funding:** This work was supported by the Major Projects of Guangdong Education Department for

## Abstract

Microctis Folium is the dried leaves of a plant (*Microcos paniculata* L.) used to improve the digestive system, alleviate diarrhoea, and relieve fever, but information regarding its chemical composition has rarely been reported. The traditional research approach of determining chemical composition has included isolating, purifying, and identifying compounds with high-cost and time-consuming processes. In this study, molecular networking (MN) and fingerprint analysis were integrated as a comprehensive approach to study the chemical composition of Microctis Folium by an ultra fast liquid chromatography-photo diode array detector-triple-time of flight-tandem mass spectrometry (UFLC-DAD-Triple TOF-MS/MS). Large numbers of mass spectrometric data were processed to identify constituents, and the identified compounds and their unknown analogues were comprehensively depicted as visualized figures comprising distinct families by MN. A validated fingerprint methodology was established to quantitatively determine compounds in Microctis Folium. Ultimately, 165 constituents were identified in Microctis Folium for the first time and the identified compounds and approximately five hundred unknown analogues were applied to create visualized figures by MN, indicating compound groups and their chemical structure analogues in Microctis Folium. The validated fingerprint methodology was indicated to be specific, repeatable, precise, and stable and was used to determine 15 batches of samples during three seasons in three districts. Furthermore, seasonal or geographic environmental influences on the chemical profile were estimated by principal coordinate analysis. The results can be used to control the quality of Microctis Folium, observe seasonal or geographic environmental influences on the chemical profiles, and provide a reference for further exploitation of potential active unknown analogues in the future.

Foundation Research and Applied Research (CN) (Grant No.2017GKZDXM010), http://edu.gd.gov. cn/. The funder had no role in study design, data collection and analysis, decision to publish, or preparation of the manuscript.

## Introduction

Microctis Folium (MF) is the dried leaves of a plant (*Microcos paniculata* L.) native to southern China. In summer and autumn, many herbal teas are decocted with these leaves in hot water to aid the digestive system, alleviate diarrhoea, and treat symptoms of fever caused by the common cold. Some studies have reported on the constituents and pharmacological effects of MF, including the anti-inflammatory effects of apigenin C-glycosides; the α-glucosidase inhibitory effects of vitexin, isovitexin and isorhamnetin 3-O-β-D-rutinoside; the antioxidative activity of epicatechin; the antitumour activities of seven alkaloids; and the hepatoprotective effect and antinociceptive, antidiarrhoeal, and antipyretic activities of MF extracts [1–8]. There are few reports about the comprehensive chemical profiles of MF, hindering its further exploitation.

Natural products extracted from living organisms, including microorganisms, plants and animals, have diverse chemical structures and promising biological activity, inspiring numerous new drugs [9–10]. The classical methods of studying chemical composition include isolating, purifying, and identifying compounds in plants. Recent technological advances in high-resolution mass spectrometry have facilitated the simultaneous identification of massive numbers of chemical structures in small quantities in complex matrices. Compared with nuclear magnetic resonance spectroscopy, which provides an extremely accurate elucidation of chemical structures, high-resolution mass spectrometry exhibits superior sensitivity and can simultaneously determine diverse compounds in several minutes, but it can also be used to identify structures with the assistance of standard or MS databases. In addition, tremendous quantities of MS/MS spectral data have made identification excessively complicated to manually analyse and have been consistently verified by standard substances or MS databases. Several free online MS databases, such as HMDB [11], KEGG [12], Metlin [13], mzCloud, and XCMS Online [14], are readily available, but they mostly emphasize data analyses of known metabolites and are seldom used to identify structures of unknown compounds. Some approaches have been reported to predict the molecular structures of unknown metabolites through MS/MS spectra, including CSI:-FingerID, the MCID MS/MS search program, and the combination of molecular networking with an in-silico MS/MS database [15]. The drawbacks of these approaches are that they have seldom been used to systematically determine the chemical profiles and quantitatively evaluate compounds, including known and unknown constituents in plants.

Recently, Global Natural Products Social Molecular Networking (GNPS; http://gnps.ucsd. edu) has been reported as a web-based mass spectrometry ecosystem that aims to serve as an open-access database platform for the storage, analysis, and sharing of raw, processed or identified tandem mass (MS/MS) spectrometry data [16]. In addition to integrating massive MS/MS databases relevant to known compounds, GNPS can enable molecular networking (MN), comparing all MS/MS spectra, visualizing spectral correlations, and detecting data sets of spectra from related molecules, even if the spectra are not matched to any known compounds. Although MN was initially introduced in metabolomics [17–19], other successful applications of MN have been reported, including isolating several bioactive compounds [20–22], revealing the cryptic chemical traits that mediate plant ecology and evolution [23], discovering and characterizing microbial natural products [24], and establishing biological omics data relationships between humans, plants and microbes [25]. The chemical profile analysis of diverse components has been rendered efficient, convenient and comprehensive by MN in plants since MN not only can be used to identify known compounds but also hints at unknown analogues. Except for constituent analysis, a validated method is required to quantitatively measure quality by multi-component analysis. Fingerprint analysis is a comprehensive quantifiable or semi-quantifiable method that can be used to evaluate the quality and stability of traditional Chinese medicine [26]. This analysis does not focus on a single compound but rather on the integral

character of constituents in the analysis of traditional Chinese medicine. Many chromatographic technologies, including thin-layer chromatography, gas chromatography, capillary electrophoresis, and high-performance liquid chromatography, can be used in fingerprint analysis [26]. Among these approaches, thin-layer chromatography is a rapid and simple method with poor resolution and low sensitivity [27]; gas chromatography is mainly used to determine volatile compounds [28]; and capillary electrophoresis has high resolution and fast analysis speed, but it is often used to analyse proteins, nucleotides, etc [29]. It is well known that high-performance liquid chromatography (HPLC) is the most common fingerprinting approach due to its high sensitivity, accuracy, precision, and repeatability as well as wide applicability to hydrophilic and lipophilic compounds [30].

In this study, 165 constituents were identified in MF for the first time, and approximately five hundred unknown analogues were demonstrated by MN. All identified constituents and their analogues were presented via MN, indicating comprehensive chemical profiles of MF. Moreover, a method was established through chromatographic fingerprint validation by HPLC and applied to evaluate the quality of 15 batches of MF collected from different regions during different seasons. Furthermore, seasonal or geographic environmental influences on the chemical profile were estimated by principal coordinate analysis (PCoA) based on MS/MS spectra. Our study provided a strategy integrating MN with fingerprint analysis to comprehensively study the chemical composition in plants. The approach can also be applied to demonstrate the chemical composition and control quality of MF, determine environmental influences, and identify unknown analogues as the basis for further exploitation in the future.

## Materials and methods

### Reagents and materials

Fifteen batches of Microctis Folium (*Microcos paniculata* L.) were collected from three districts of Guangdong Province, China, in three seasons (three batches from Suixi in summer; three batches from Suixi in winter; one batch from Yangchun in summer; four batches from Yangchun in autumn; three batches from Yangxi in autumn; and one batch from Yangxi in winter), as shown in S1 Table. All samples were deposited in the Guangdong Key Laboratory of Plant Resources and identified by chief pharmacist Liwei Yang (Guangdong Institute for Food and Drug Control, Guangzhou, China). Vitexin (111687–200602, purity: 99.9%) and Microctis Folium reference herb (121218–201104) were purchased from the National Institute for Control of Biological and Pharmaceutical Products of China (Beijing, China). Methanol of HPLC grade (Honeywell, Morris Plains, NJ, USA) and formic acid of HPLC grade (Sigma-Aldrich, Guangzhou, China) were used. All water used was distilled and further purified by a Milli-Q system (Millipore, Milford, MA, USA). Other reagents used in the experiment were analytical grade.

### Preparation of the standard solution and sample

An appropriate amount of standard (vitexin) was accurately weighed and dissolved in 70% methanol (methanol: water = 7:3, v:v) to obtain the standard solution (20μg mL$^{-1}$). After 2.5 g of sample was accurately weighed in a 100 mL glass-stoppered conical flask, 50 mL of 70% methanol (methanol: water = 7:3, v:v) was added. The filled flask was weighed with a precision of ±0.01 g, sonicated for 30 min (250 W, 33 kHz) twice, allowed to cool, and adjusted to the initial weight with 70% methanol as needed. Then, the solution was filtered with a membrane filter (0.45 μm) to collect the successive filtrate. The filtrate was used as the sample solution.

## Analysis by UFLC-DAD-Triple TOF-MS/MS

Analysis was performed with a Shimadzu XR ultra fast liquid chromatography (UFLC) (Shimadzu Corp., Kyoto, Japan) equipped with an in-line degasser, a binary pump, an autosampler, a column oven, and a photodiode array detector (DAD). Separation was carried out on an Agilent Zorbax Eclipse Plus C18 column (4.6mm×250mm, 5μm) at 40˚C. The mobile phase consisted of 0.1% formic acid (v/v) in methanol (A) and 0.1% formic acid (v/v) in water (B) using a linear gradient from 25 to 40% A (0–20 min), 40 to 60% A (20–70 min), 60% to 90% A (70–71 min), 90% A (71–75 min), 90% to 25% A (75–76 min), and 25% A (76–80 min). The injected volume was 10 μL, with the flow rate kept at 0.3 mL/min. The DAD scanned from 190 to 400 nm. The optimal wavelength was 339 nm.

Detections were performed by an AB SCIEX 5600 plus triple quadrupole time-of-flight mass spectrometry (Triple TOF-MS/MS) (AB Sciex, Foster City, CA, USA) equipped with electrospray ionization (ESI) source and Peak View analysis software (Version 2.1, AB Sciex, Shanghai, China). The TOF-MS operated in full scan mode, and the mass range was set to $m/z$ 100–1200 in both the positive and negative ion modes. MS/MS data were acquired by the information-dependent acquisition (IDA) mode. The conditions of the mass spectrometry were as follows: ion source gas1, 55 psi; ion source gas2, 55 psi; curtain gas, 35 psi; temperature, 550˚C; ion spray voltage floating, 5500 V (positive) or −4500 V (negative); collision energy, 40 V; collision energy spread, 20 V; and declustering potential, 80 V. Nitrogen was used as the nebulizer and auxiliary gas.

## Molecular Networking (MN)

After all MS data files (wiff type) were converted to the required formats (mzxml type) by Proteo Wizard Software (Version: 3.0.11781), they were uploaded to the GNPS website (http://gnps.ucsd.edu) by File Zilla Software (Version: 3.32.0). A MN was established using the online workflow of GNPS (http://gnps.ucsd.edu). Through the "View Spectra Families" function, the MN could be visualized at a higher level, in which each MS/MS spectrum was represented as a node and spectrum-to-spectrum relatedness as edges between nodes. The data were clustered with MS-Cluster with a parent MS tolerance of 0.2 Da and an MS/MS fragment ion tolerance of 0.2 Da to create consensus spectra containing more than 2 spectra. The relatedness was filtered to have a cosine score above 0.7 and more than 6 matched peaks in the MS/MS spectrum. After the parameters were appropriately set, the spectra in the network were searched against the NIH Natural Products library available in GNPS to identify constituents that were not reported in MF. Furthermore, other constituents in MN were listed, whose spectra could not be the same but were analogous to those in the NIH Natural Products library. To evaluate the influence of season and geography, the MF samples were divided into different groups according to seasons or districts; then, the PCoA plots were processed on the results page by clicking "View Emporer PCoA Plot in GNPS".

## Fingerprint analysis

Methodology validation of fingerprint analysis was established according to the Guidelines for High Performance Liquid Chromatography in Chinese Pharmacopoeia (Chinese Pharmacopoeia Commission, version 2015, part IV) and the Technical Guideline for Experimental Study of Chromatogram Fingerprint of Traditional Chinese Medicine Injection (Chinese Pharmacopoeia Commission, 2002).

**Specificity.** Equal volumes of the blank solvent, standard solution (vitexin), Microctis Folium reference herb, and sample solution were separately injected into the chromatograph, and the chromatograms were recorded.

**Precision.** The repeatability was evaluated by injecting six sample solutions of the same batch into the instrument to calculate their relative retention time (RRT), relative peak area

(RPA), and similarities. The intermediate precision was assessed by preparing sample solutions independently in duplicate with the same sample by two operators, on two days, and injecting the solutions into different instruments to calculate their RRT, RPA, and similarities.

**Stability.** The stability of the sample solution was investigated. This process was carried out by comparing the RRT, RPA, and similarities in the chromatographs of the same sample solution after being stored at room temperature for different times (0, 4, 8, 12, 24, and 48 hours).

**Ruggedness.** The ruggedness was evaluated by examining the peak stability, with small variations in procedural parameters on different columns (Welch Ultimate XB-C18, 4.6×250mm, 5μm; Agilent Zorbax Eclipse Plus C18, 4.6×250 mm, 5μm; Elite Hypersil ODS2-C18, 4.6×250 mm, 5μm); the RRT, RPA, and similarities of each characteristic peak were investigated.

The UFLC-DAD spectra were used to calculate similarities with the Similarity Evaluation System for Chromatographic Fingerprint of Tradition Chinese Medicine Software (Version 2012.1, Chinese Pharmacopoeia Commission). The similarities in methodology validation should be between 0.90 and 1.00.

## Results and discussion

### Identification of compounds with GNPS

With the development of technology, high-resolution mass spectrometry can identify constituents by information from standard substances or MS databases, including exact molecular weight, exact mass and intensity of fragment ions. Because there is little knowledge about constituents in MF, the MS spectra were continuously searched against all GNPS spectral knowledge to elucidate the chemical profile. After matching the MS spectra of GNPS, 168 compounds were identified in MF, including 81 flavonoids, 21 alkaloids, 18 lignins, 11 terpenoids, 7 polyols, 5 lignans, 4 phenols, 3 coumarins, 3 steroids, and 15 other groups (shown in S1 Fig and S2 Table). The identified details of the fragment ion characteristics are shown in S3 Table. Except for vitexin, epicatechin and isorhamnetin 3-O-β-D-rutinoside [1, 4–5], 165 compounds were reported in MF for the first time. The results of the chemical profiles contribute to quality control and mechanistic research of constituents in MF.

### Molecular Networking (MN)

To present the chemical profile of MF, the acquired compounds and their analogues were comprehensively depicted as MNs, comprising distinct families (shown in Fig 1, S2 and S3 Figs, and S4 Table). After families were processed with Cytoscape software, their nodes were clustered into several special circles, indicating constituents groups and their refined structural differences. The analogical mass-spectrometric fragment characteristics produced several clusters of nodes, presenting similar chemical structures. In addition, many unknown analogues could be inferred based on similar structural characteristics from the known compounds in the same families of MNs, and their exact molecular weights are presented in Fig 1. Abundant flavonoids were reported in MF, including flavonols, flavones, isoflavones, flavans, anthocyanidins, and xanthones. Most flavonols belonged to families Pos-01, Pos-10, and Neg-10 (Fig 1 and S2 Fig). The reason was that the major fragmentation pathway of flavonol glycosides focused on analogue retro-Diels-Alder reactions and the loss of saccharides in structures despite the different mass-to-charge ratios of the product ions (S3 and S4 Tables). Although the structures of compounds in family Pos-01 all contained 4′, 5, 7-trihydroxyflavone glycosides, slight functional group differences at other sites produced clusters of three separate circles. It appeared that the clusters of MNs depended on structural similarity rather than the

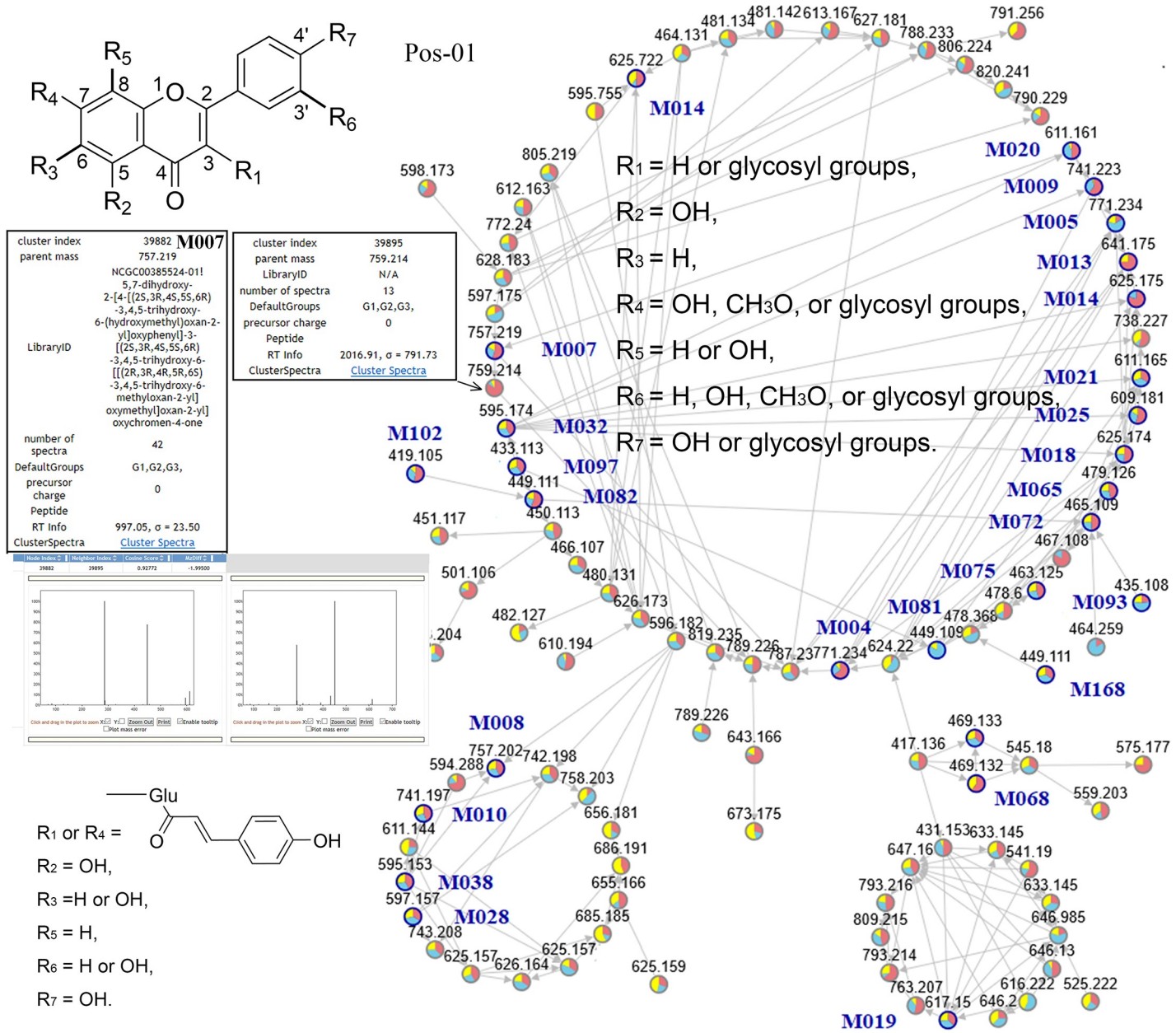

**Fig 1. Molecular Networking (MN) visualization of family Pos-01.** Each node is represented as a compound's MS/MS spectrum because all retention times of nodes are different. Blue nodes are represented as spectra of identified constituents, which include noted compounds' names, and grey nodes are represented as spectra of their unknown analogues. Taking M007 and its analogue as an example, the information of identified constituents and their unknown analogues can be presented as noted on the left, including the name, parent mass, retention time, cosine score, *m/z* error, and mass fragmentation spectrum. Each coloured wedge of a node pie is the proportion of the spectral counts derived from each respective season (red: summer, blue: autumn, yellow: winner). Each edge between nodes is represented as a spectrum-to-spectrum similarity cosine score above 0.7.

values of exact mass of the parent ion and product ion. For example, all identified known compounds contained para-coumaryl glucosides in the second larger circle of Pos-01, including the lignin M028. Then, their analogues could be inferred to have similar structural characteristics, containing 4′, 5, 7-trihydroxyflavone glycosides and/or para-coumaryl glucosides in the same circle. Since the M068 structure was partially similar to that of 4′, 5, 7-trihydroxyflavone,

it belonged to the family Pos-01. In addition, the structures of compounds in family Pos-10 included 4′, 5, 7-tetrahydroxyflavone without any glycosyl group.

The MNs of flavones mainly included the family Pos-02, Neg-04, Neg-07, and Neg-13 because the major fragmentation characteristics of flavone glycosides were from similar dehydration processes, retro-Diels-Alder reactions, the loss of saccharides, and the fracture of the saccharide ring (Fig 2, S3 Fig, and S3 and S4 Tables). In the largest circle of family Pos-02 and family Neg-04, the structural character of flavones was that two glycosyls attached to 4′, 5, 6, 7, 8-pentahydroxyflavone or 4′, 5, 6, 7, 8-pentahydroxyisoflavone. The second largest circle of family Pos-02 was composed of compounds including two glycosyl groups attached to 4′, 5, 6, 7-tetrahydroxyflavone or 4′, 5, 7, 8-tetrahydroxyflavone. Most compounds were 4′, 5, 6, 7-tetrahydroxyflavone or 4′, 5, 7, 8-tetrahydroxyflavone attached to only one glycosyl moiety in the third largest circle of Pos-02, which were also in the largest circle of family Neg-07. In family Neg-13, compounds showed the major structures of 4', 5, 7-trihydroxyflavanone or 4', 5, 7-trihydroxyflavone without any glycosyl units.

The flavans were largely distributed to families Pos-04, Pos-12, Neg-01, Neg-06, Neg-11, Neg-12, Neg-13, and Neg-21 (S3 and S4 Figs and S4 Table) since the main fragmentation characteristics of flavan glycosides were from similar dehydration processes, retro-Diels-Alder reactions, the loss of saccharides, the removal of carbon monoxide, and the fracture of the saccharide ring (S3 Table). Although there were many constituents (nodes) in the corresponding families (MNs), only M043 was in the Pos-04 and Neg-01 families, and M029 was in Neg-06, indicating that M043 and M029 might have diverse analogues in MF. M043 and M044 both contained two epicatechin units, forming family Pos-12 with their analogues. In Neg-12, compounds M029, M039, and M040 have the diglycosyl moiety attached to carbon 7 at the A ring of 4', 5, 7-trihydroxyflavanone. Some flavans and flavones without any glycosyl groups, including M131, M133, M140, and M141, belonged to Neg-13.

Other compounds in MF, including alkaloids, coumarins, lignans, phenols, polyols, steroids, and terpenoids, had a large variety of structures and formed many small MNs with their analogues, although they belonged to the same class (S3 and S4 Tables).

In the development of new drugs from natural resources, researchers are keen to elucidate the initial chemical profiles of complicated mixtures for further studies. Furthermore, if some known constituents have been reported in other plants, then research of their unknown analogues can be facilitated. With comprehensive information pertaining to MS spectra, the 168 compounds and approximately five hundred unknown analogues were presented in MNs. The results could aid researchers in expanding their knowledge of MF's chemical profiles and provide some clues for further exploitation of promising unknown analogues.

## Principal Coordinate Analysis (PCoA)

In the prevailing view, seasonal or environmental factors readily cause chemical composition variations in plants [9]. There were too few peaks in the fingerprinting figures to evaluate the influences of UFLC-DAD. Therefore, abundant MS data of hundreds of constituents were processed by PCoA to estimate the influence of season and districts (shown in Fig 3).

The results demonstrated that the samples picked during the same season in the same district, resembled each other, as expected. However, the samples gathered in the same season were readily clustered, as shown in A of Fig 3. For the samples from Suixi, including S-01, S-02, S-03, S-12, S-13, and S-14, summer and winter made them slightly distant in B of Fig 2. The same occurred for the samples from Yangchun, including S-05, S-06, S-07, and S-15 collected in autumn and winter. However, the samples in Yangxi were indistinguishable even though they were picked in summer and autumn. It appears that the influence of reasons was more obvious than that of districts for the chemical composition of MF.

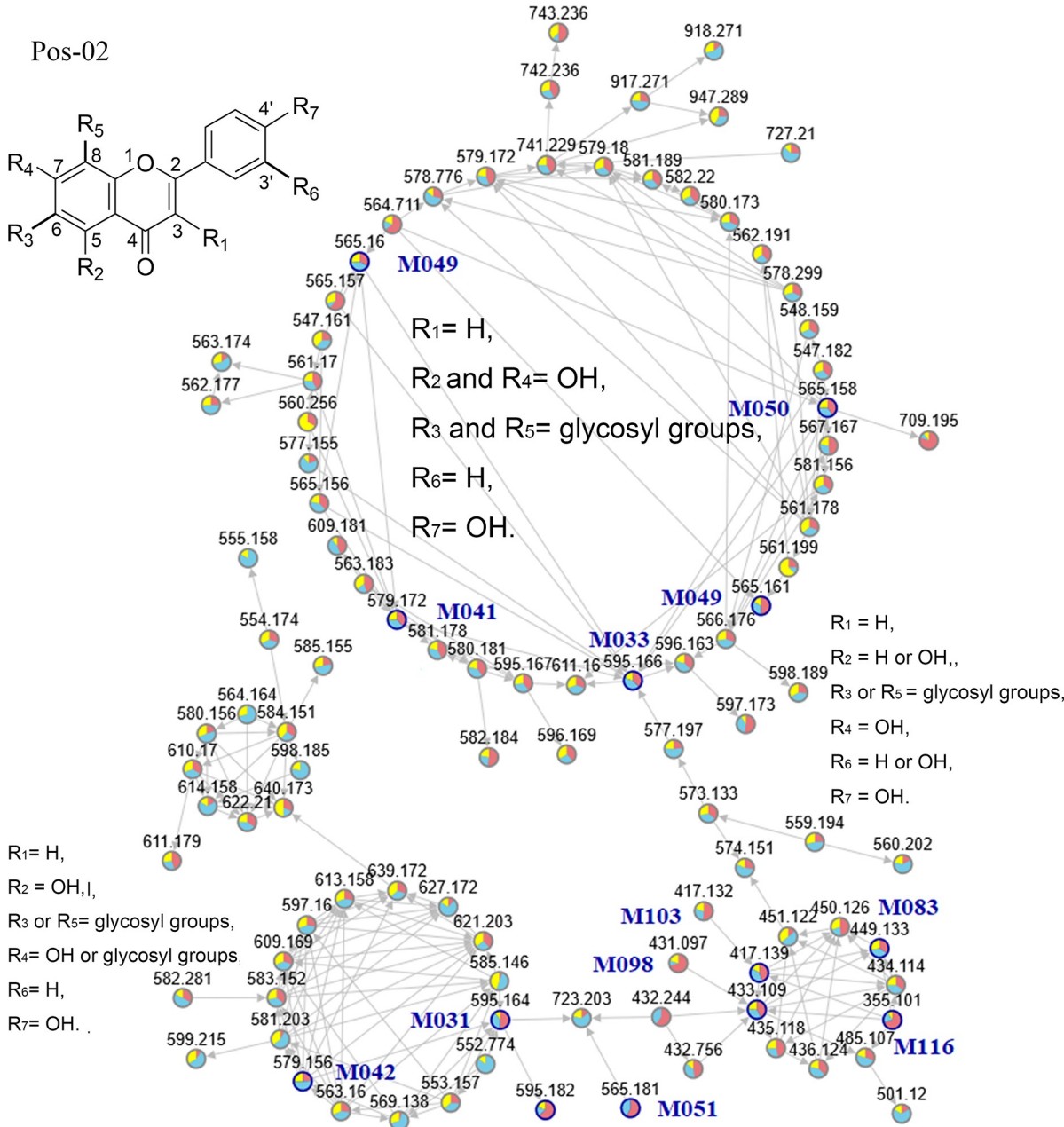

**Fig 2. Molecular Networking (MN) visualization of family Pos-02.** Each node is represented as a compound's MS/MS spectrum because all retention times of nodes are different. Blue nodes are represented as spectra of identified constituents, which include noted compounds' names, and grey nodes are represented as spectra of their unknown analogues. Each coloured wedge of a node pie is the proportion of the spectral counts derived from each respective season (red: summer, blue: autumn, yellow: winner). Each edge between nodes is represented as a spectrum-to-spectrum similarity cosine score above 0.7.

## Fingerprint analysis

**Optimization of extraction and chromatographic conditions.** Different proportions of methanol water solution (50%, 70%, and 100%) were tested to extract maximum constituents. The optimal extraction method was ultrasonic extraction with a methanol water solution (70%, v: v). The determination wavelength was chosen as 339 nm after the chromatograms

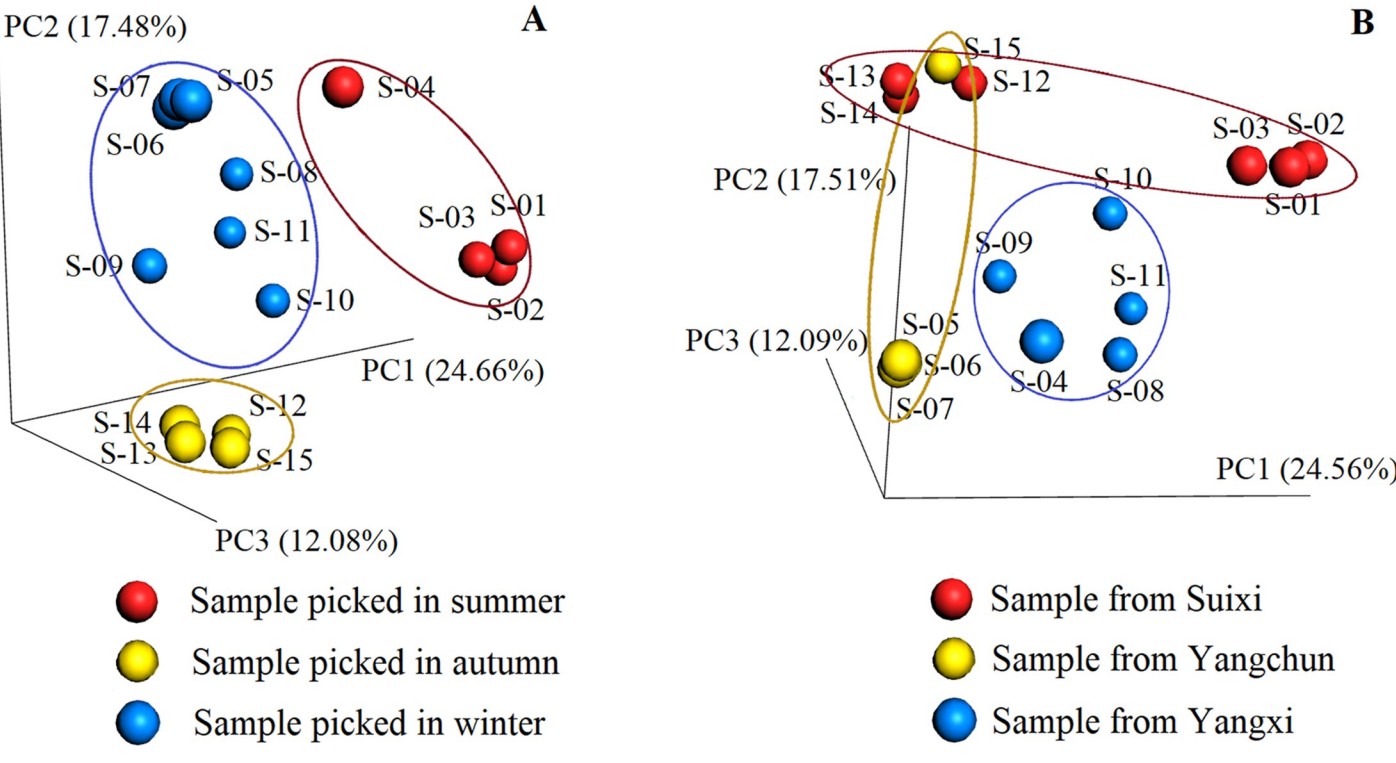

**Fig 3.** Principal coordinate analysis (PCoA) according to season (A) and geographical environment (B).

were scanned by DAD. The gradient elution program was optimized to acquire higher resolution for chromatographic peaks, and then, three kinds of columns were used to test its feasibility. The seven peaks were chosen as common peaks in fingerprint analysis due to their areas, resolutions, and purities.

**Methodology validation.** *Specificity*. The integration peaks in the chromatogram of the sample solution corresponded in time to the peaks in the chromatogram of the MF reference herb solution and the standard solution. No such peaks of that retention time appeared in the chromatogram of the solvent (S4 Fig).

*Precision*. In the repeatability and intermediate precision tests, all RSDs of the RRT and RPA were below 6.00%, and all similarities were between 0.90 and 1.00, indicating high precision (Table 1).

*Stability*. The RSDs of the RRT and RPA were all below 3.16%, and the similarities were between 0.90 and 1.00 over 48 hours. The seven common peaks were presented to be stable for sample solutions over 48 hours (Table 1).

*Ruggedness*. After three kinds of columns were used to evaluate the ruggedness, the RSDs of the RRT and RPA were all below 7.46%, and the similarities were between 0.90 and 1.00. The method showed good ruggedness on different columns (Table 1).**Sample fingerprint analysis.** After methodology validation, the similarities between 15 batches of MF samples and the reference fingerprint were calculated with the Similarity Evaluation System for Chromatographic Fingerprint of Tradition Chinese Medicine Software (Version 2012.1, Chinese Pharmacopoeia Commission). The content of vitexin was determined as shown in Table 2 according to the regulation of MF in the Chinese Pharmacopoeia (Version 2015, Part I, Chinese Pharmacopoeia Commission). The fingerprint chromatogram and the similarities are presented in Fig 4

**Table 1. The results of repeatability, intermediate precision, stability, and ruggedness for the seven common peaks.**

| Peak No. | Repeatability (n = 6) | | Intermediate precision (n = 4) | | Stability (n = 6) | | Ruggedness (n = 3) | |
|---|---|---|---|---|---|---|---|---|
| | RSD of RRT (%) | RSD of RPA (%) | RSD of RRT (%) | RSD of RPA (%) | RSD of RRT (%) | RSD of RPA (%) | RSD of RRT (%) | RSD of RPA (%) |
| 1 | 0.29 | 0.88 | 0.55 | 6.00 | 0.19 | 2.52 | 5.03 | 4.14 |
| 2 (S) | 0.00 | 0.00 | 0.00 | 0.00 | 0.00 | 0.00 | 0.00 | 0.00 |
| 3 | 0.10 | 0.18 | 0.32 | 0.16 | 0.13 | 0.17 | 2.42 | 4.75 |
| 4 | 0.19 | 1.83 | 0.74 | 4.88 | 0.25 | 2.65 | 0.35 | 6.11 |
| 5 | 0.20 | 0.50 | 0.28 | 2.15 | 0.19 | 1.11 | 1.23 | 7.41 |
| 6 | 0.21 | 0.14 | 0.42 | 0.60 | 0.20 | 0.54 | 0.80 | 3.59 |
| 7 | 0.37 | 0.13 | 2.06 | 2.39 | 0.42 | 3.16 | 6.75 | 7.46 |

RRT: Relative retention time.

RPA: Relative peak area.

and Table 2. All similarities were between 0.90 and 1.00, with a vitexin content above 0.040%, indicating a relatively consistent chromatogram by UFLC-DAD. The results presented a precise, repeatable, and stable method to control the quality of MF.

## Conclusions

In this study, MN and fingerprint analysis were combined as a strategy to study the chemical composition in MF, which focused on integral characteristics of the identified constituents and their analogues. Of 168 identified compounds, 165 constituents were identified in MF for the first time, and approximately five hundred unknown analogues were presented by MN, which aided in systematizing the MS/MS spectra data and determining the chemical profile of MF. PCOA was applied to estimate the influence of seasonal and geographical environment variations. Fingerprint analysis was used to quantitatively analyse common peaks according to the overall appearance of the chromatogram. This strategy can be used to understand the

**Table 2. The similarities of 15 batches of MF in fingerprint analysis.**

| Batch No. | Similarity | Content of vitexin (%) |
|---|---|---|
| S1 | 0.998 | 0.0687 |
| S2 | 0.985 | 0.0650 |
| S3 | 0.987 | 0.0683 |
| S4 | 0.978 | 0.0620 |
| S5 | 0.966 | 0.0477 |
| S6 | 0.966 | 0.0593 |
| S7 | 0.959 | 0.0488 |
| S8 | 0.986 | 0.0956 |
| S9 | 0.988 | 0.0658 |
| S10 | 0.983 | 0.0837 |
| S11 | 0.946 | 0.1597 |
| S12 | 0.996 | 0.0538 |
| S13 | 0.998 | 0.0625 |
| S14 | 0.997 | 0.0588 |
| S15 | 0.998 | 0.0489 |

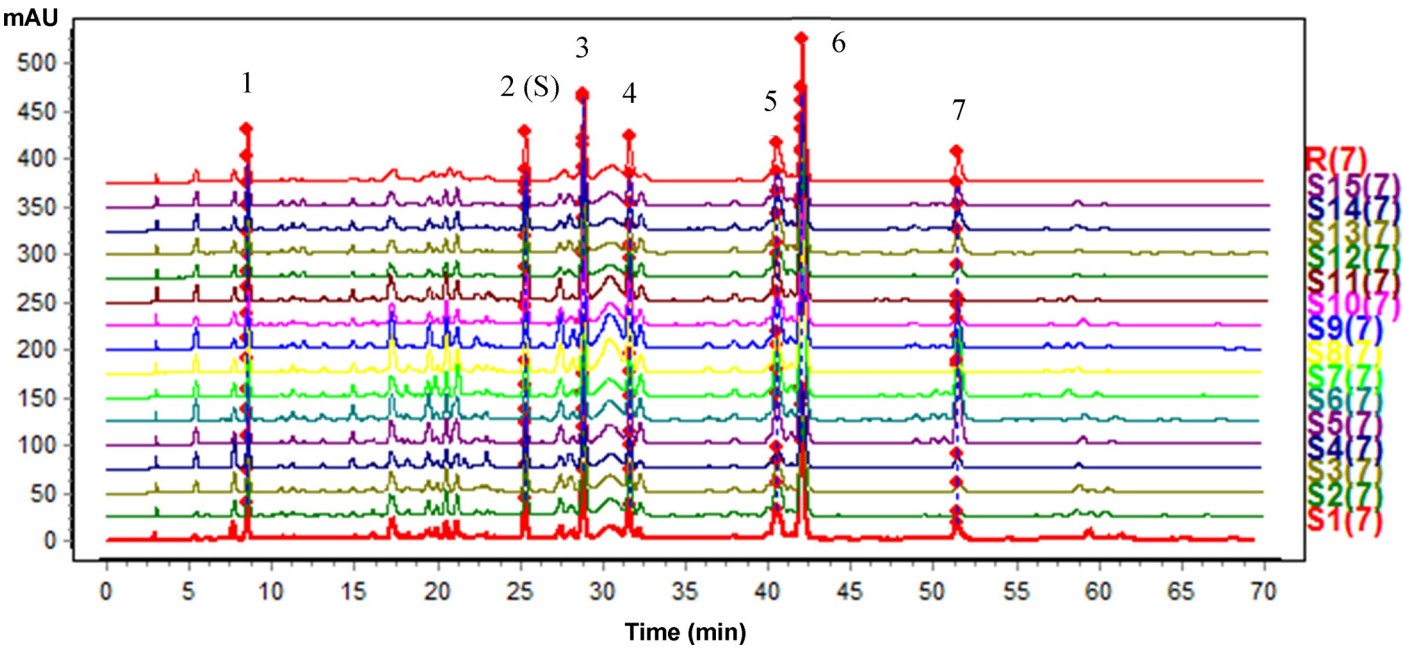

**Fig 4. The fingerprint chromatogram of 15 batches of MF.** R: Reference fingerprint, 2 (S): Vitexin.

chemical profiles, control the quality of MF, and contribute to discovering unknown analogues for further exploitation in the future.

## Supporting information

**S1 Fig. The molecular structures of the identified compounds.**
(TIF)

**S2 Fig. Molecular networking (MN) visualization of families Pos-10 and Neg-10.** Each node is represented as a compound's MS/MS spectrum because all retention times of nodes are different. Blue nodes are represented as spectra of identified constituents, which include noted compounds' names, and grey nodes are represented as spectra of their unknown analogues. Each coloured wedge of a node pie is the proportion of the spectral counts derived from each respective season (red: summer, blue: autumn, yellow: winner). Each edge between nodes is represented as a spectrum-to-spectrum similarity cosine score above 0.7.
(TIF)

**S3 Fig. Molecular networking (MN) visualization of families Neg-04, Neg-07, and Neg-13.** Each node is represented as a compound's MS/MS spectrum because all retention times of nodes are different. Blue nodes are represented as spectra of identified constituents, which include noted compounds' names, and grey nodes are represented as spectra of their unknown analogues. Each coloured wedge of a node pie is the proportion of the spectral counts derived from each respective season (red: summer, blue: autumn, yellow: winner). Each edge between nodes is represented as a spectrum-to-spectrum similarity cosine score above 0.7.
(TIF)

**S4 Fig. Molecular networking (MN) visualization of families Pos-04, Pos-12, Neg-01, and Neg-11.** Each node is represented as a compound's MS/MS spectrum because all retention times of nodes are different. Blue nodes are represented as spectra of identified constituents,

which include noted compounds' names, and grey nodes are represented as spectra of their unknown analogues. Each coloured wedge of a node pie is the proportion of the spectral counts derived from each respective season (red: summer, blue: autumn, yellow: winner). Each edge between nodes is represented as a spectrum-to-spectrum similarity cosine score above 0.7.

(TIF)

**S5 Fig. Specificity of methodology validation of MF fingerprint.** A: Standard solution of vitexin, B: solvent, C: Solution of MF reference herb, D: Sample solution of MF, 2 (S): Vitexin.

(TIF)

**S1 Table. Sample information.**
(DOCX)

**S2 Table. Identified compounds' information.**
(DOCX)

**S3 Table. Identification of compoundsin MF.**
(DOCX)

**S4 Table. Molecular networking families' information.**
(DOCX)

## Author Contributions

**Conceptualization:** Yang Bai, Qiang Jia, Hongliang Yao.

**Data curation:** Yang Bai, Weiwei Su, Zenghao Yan, Hongliang Yao.

**Formal analysis:** Yang Bai, Qiang Jia.

**Funding acquisition:** Hongliang Yao.

**Investigation:** Yang Bai, Qiang Jia, Weiwei Su, Zenghao Yan, Hongliang Yao.

**Methodology:** Yang Bai, Qiang Jia, Weiwei Su, Zenghao Yan, Wenhui Situ, Xiang He, Wei Peng, Hongliang Yao.

**Project administration:** Yang Bai, Weiwei Su, Hongliang Yao.

**Resources:** Yang Bai, Qiang Jia, Weiwei Su, Hongliang Yao.

**Supervision:** Weiwei Su, Hongliang Yao.

**Validation:** Yang Bai, Qiang Jia, Weiwei Su, Zenghao Yan, Wenhui Situ, Xiang He, Wei Peng, Hongliang Yao.

**Visualization:** Yang Bai, Qiang Jia.

**Writing – original draft:** Yang Bai, Qiang Jia, Hongliang Yao.

**Writing – review & editing:** Yang Bai, Qiang Jia, Zenghao Yan, Hongliang Yao.

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
