## [Decision Letter · Decision Letter 0]

5 Mar 2020

PONE-D-19-30659

Integration of molecular networking and fingerprint analysis for studying constituents in Microctis Folium

PLOS ONE

Dear Yang Bai,

Thank you for submitting your manuscript to PLOS ONE. After careful consideration, we feel that it has merit but does not fully meet PLOS ONE’s publication criteria as it currently stands. Therefore, we invite you to submit a revised version of the manuscript that addresses the points raised during the review process.

We would appreciate receiving your revised manuscript by 21 March. To enhance the reproducibility of your results, we recommend that if applicable you deposit your laboratory protocols in protocols.io, where a protocol can be assigned its own identifier (DOI) such that it can be cited independently in the future. For instructions see: http://journals.plos.org/plosone/s/submission-guidelines#loc-laboratory-protocols

We look forward to receiving your revised manuscript.

Kind regards,

Tommaso Lomonaco, Ph.D

Academic Editor

PLOS ONE

Additional Editor Comments (if provided):

Dear authors, please address all the comments of the reviewer.

Regards,

Tommaso Lomonaco

Journal Requirements:

Reviewers' comments:

Reviewer's Responses to Questions

**Comments to the Author**

1. Is the manuscript technically sound, and do the data support the conclusions?

Reviewer #1: Yes

2. Has the statistical analysis been performed appropriately and rigorously? 

Reviewer #1: Yes

3. Have the authors made all data underlying the findings in their manuscript fully available?

Reviewer #1: Yes

4. Is the manuscript presented in an intelligible fashion and written in standard English?

Reviewer #1: Yes

5. Review Comments to the Author

Reviewer #1: 1. Author identified 165 constituents in source material MF using MN methodology, Is it exact identification without overlapping in the data used.

2. Author extracted MF with 70% methanol and used cured extract itself. However, it would be better if the extract was fractionated with different solvents to reduce complexity and to say exactly which portion and constituents are responsible for its medicinal use.

3. Author has collected the source material from three districts of Guangdong province in three seasons to assess the geographical and seasonal influence on constituents. But these districts may not vary much geographically. It would be better if author collect MF from varied places of china or world.

6. PLOS authors have the option to publish the peer review history of their article (what does this mean?). If published, this will include your full peer review and any attached files.

Reviewer #1: No

---

## [Author Response · Author response to Decision Letter 0]

20 Mar 2020

1. Is the manuscript technically sound, and do the data support the conclusions?

Reviewer #1: Yes

Response:

Thanks for your comment and approval.

2. Has the statistical analysis been performed appropriately and rigorously? 

Reviewer #1: Yes

Response:

Thanks for your comment and approval.

3. Have the authors made all data underlying the findings in their manuscript fully available?

Reviewer #1: Yes

Response:

Thanks for your comment and approval.

4. Is the manuscript presented in an intelligible fashion and written in standard English?

Reviewer #1: Yes

Response:

Thanks for your comment and approval.

5. Review Comments to the Author

Reviewer #1: 

1. Author identified 165 constituents in source material MF using MN methodology, Is it exact identification without overlapping in the data used.

Response:

Yes. The identified constituent detailed information was shown in S1 Fig and S2 and S3 Tables, including name, molecular structure, chemical formula, rentention time, observed exact mass, mass error, and main fragment ions. In line 220-231, our results were compared with the reported results in literature about Microctis Folium. All information presented exact identification results without overlapping in the data used.

2. Author extracted MF with 70% methanol and used cured extract itself. However, it would be better if the extract was fractionated with different solvents to reduce complexity and to say exactly which portion and constituents are responsible for its medicinal use.

Response:

In line 48-58, MF showed the effects of aiding the digestive system, alleviating diarrhoea, and treating symptoms of fever caused by the common cold when it is decocted with hot water. So, in this study, we preferred to extract hydrophilic constituents in MF, which were extracted with 50%, 70% and 100% methanol. 

3. Author has collected the source material from three districts of Guangdong province in three seasons to assess the geographical and seasonal influence on constituents. But these districts may not vary much geographically. It would be better if author collect MF from varied places of china or world.

Response:

Initially, we tried to collect MF from more districts. However, we found MF was hardly available from many places of world by local herbal markets or online shopping system. Furthermore, some batches of MF failed to provide detailed location information, including latitude, longitude, and altitude, and couldn’t be used in this study, which hindered MF collection from varied places of china or world.

6. PLOS authors have the option to publish the peer review history of their article (what does this mean?). If published, this will include your full peer review and any attached files.

Do you want your identity to be public for this peer review? For information about this choice, including consent withdrawal, please see our Privacy Policy.

Reviewer #1: No

Response:

We respect reviewer’s choice.

Dear Reviewer,

Thank you for your careful comments concerning our manuscript entitled “Integration of molecular networking and fingerprint analysis for studying constituents in Microctis Folium” [PONE-D-19-30659].

We have studied comments carefully and responded to reviewer which we hope meet with approval. Revised portion are marked using the "Track Changes" function in Microsoft Word and the responses are followed the reviewer’s comments.

We appreciate your kind advice and careful work. If anything is not clear or in case of whatever question in the manuscript, please do not hesitate to contact us.

---

## [Decision Letter · Decision Letter 1]

20 May 2020

PONE-D-19-30659R1

Integration of molecular networking and fingerprint analysis for studying constituents in Microctis Folium

PLOS ONE

Dear Dr. Bai,

Thank you for submitting your manuscript to PLOS ONE. After careful consideration, we feel that it has merit but does not fully meet PLOS ONE’s publication criteria as it currently stands. Therefore, we invite you to submit a revised version of the manuscript that addresses the points raised during the review process.

The internal editorial staff has assessed your submission, and we have concerns about the grammar, usage, and overall readability of the manuscript. We therefore request that you revise the text to fix the grammatical errors and improve the overall readability of the text in order to meet PLOS ONE publication criteria requiring that the language in submitted articles must be clear, correct, and unambiguous. We suggest you have a fluent, preferably native, English-language speaker thoroughly copyedit your manuscript for language usage, spelling, and grammar. If you do not know anyone who can do this, you may wish to consider employing a professional scientific editing service. Whilst you may use any professional scientific editing service of your choice, PLOS has partnered with both American Journal Experts (AJE) to provide discounted services to PLOS authors. AJE has extensive experience helping authors meet PLOS guidelines and can provide language editing, translation, manuscript formatting, and figure formatting to ensure your manuscript meets our submission guidelines. If the PLOS editorial team finds any language issues in text that AJE has edited, AJE will re-edit the text for free. To take advantage of this special partnership, use the following link: https://www.aje.com/go/plos/. Please note that having the manuscript copyedited by AJE or any other editing services does not guarantee acceptance for publication.

Please note that PLOS ONE does not copyedit accepted manuscripts and that one of our criteria for publication is that articles must be presented in an intelligible fashion and written in clear, correct, and unambiguous English (http://www.plosone.org/static/publication#language). 

We would appreciate receiving your revised manuscript by Jul 04 2020 11:59PM. To enhance the reproducibility of your results, we recommend that if applicable you deposit your laboratory protocols in protocols.io, where a protocol can be assigned its own identifier (DOI) such that it can be cited independently in the future. For instructions see: http://journals.plos.org/plosone/s/submission-guidelines#loc-laboratory-protocols

We look forward to receiving your revised manuscript.

Kind regards,

Tommaso Lomonaco, Ph.D

Academic Editor

PLOS ONE

and 

Vanessa Carels

Staff Editor

PLOS ONE

Reviewers' comments:

Reviewer's Responses to Questions

**Comments to the Author**

1. If the authors have adequately addressed your comments raised in a previous round of review and you feel that this manuscript is now acceptable for publication, you may indicate that here to bypass the “Comments to the Author” section, enter your conflict of interest statement in the “Confidential to Editor” section, and submit your "Accept" recommendation.

Reviewer #1: All comments have been addressed

2. Is the manuscript technically sound, and do the data support the conclusions?

Reviewer #1: Yes

3. Has the statistical analysis been performed appropriately and rigorously? 

Reviewer #1: Yes

4. Have the authors made all data underlying the findings in their manuscript fully available?

Reviewer #1: Yes

5. Is the manuscript presented in an intelligible fashion and written in standard English?

Reviewer #1: Yes

6. Review Comments to the Author

Reviewer #1: Author answered all the questions raised by me while reviewing the manuscript, satisfactorily. No further comments on the manuscript.

7. PLOS authors have the option to publish the peer review history of their article (what does this mean?). If published, this will include your full peer review and any attached files.

Reviewer #1: No

---

## [Author Response · Author response to Decision Letter 1]

5 Jun 2020

Dear Dr.Tommaso Lomonaco,

 Thanks for your careful comments concerning our manuscript entitled “Integration of molecular networking and fingerprint analysis for studying constituents in Microctis Folium” (PONE-D-19-30659R1).

 We have submitted the manuscript to American Journal Experts (AJE) for English editing. Revised portion are marked using the "Track Changes" function in Microsoft Word.

 We appreciate your kind advice and careful work. If anything is not clear or in case of whatever question in the manuscript, please do not hesitate to contact us.

Best wishes

Yang Bai

---

## [Editor Report · Decision Letter 2]

18 Jun 2020

Integration of molecular networking and fingerprint analysis for studying constituents in Microctis Folium

PONE-D-19-30659R2

Dear Dr. Yang Bai,

We’re pleased to inform you that your manuscript has been judged scientifically suitable for publication and will be formally accepted for publication once it meets all outstanding technical requirements.

Kind regards,

Tommaso Lomonaco, Ph.D

Academic Editor

PLOS ONE

Additional Editor Comments:

Dear Authors, the english level of the manuscript is significantly improve thus the article can be published in PlosOne.

Regards,

Tommaso Lomonaco

---

## [Editor Report · Acceptance letter]

23 Jun 2020

PONE-D-19-30659R2 

Integration of molecular networking and fingerprint analysis for studying constituents in Microctis Folium 

Dear Dr. Bai:

I'm pleased to inform you that your manuscript has been deemed suitable for publication in PLOS ONE. Congratulations! Your manuscript is now with our production department. 

Kind regards, 

on behalf of

Dr. Tommaso Lomonaco 

Academic Editor

PLOS ONE